# Applying the Health Stigma and Discrimination Framework to psychosis stigma in Malawi

Melissa A. Stockton[1]*, Jack Kramer[2], Joshua Chienda[3], Abigail M. Morrison[4], Harriet Akello Tikhiwa[2], Griffin Sansbury[5], Alex Zumazuma[3], Hillary Mortensen[1], Mwawi Ng'oma[6], Patrick Nyirongo[3], Isaac Mtonga[3], Jackson Devadas[4], Bonginkosi Chiliza[7], Anthony Peter Sefasi[8], Patani Mhango[9], Bradley N. Gaynes[1,10], Brian W. Pence[1], Kazione Kulisewa[3]

1 Department of Epidemiology, Gillings School of Global Public Health, University of North Carolina at Chapel Hill, Chapel Hill, North Carolina, United States of America, 2 University of North Carolina Project-Malawi, Lilongwe, Malawi, 3 Department of Psychiatry and Mental Health, Kamuzu University of Health Sciences, Blantyre, Malawi, 4 Department of Health Behavior, Gillings School of Global Public Health, University of North Carolina at Chapel Hill, Chapel Hill, North Carolina, United States of America, 5 Miller School of Medicine, University of Miami, Miami, Florida, United States of America, 6 St. John of God Hospitaller Services, Lilongwe, Malawi, 7 Department of Psychiatry, Nelson R Mandela School of Clinical Medicine, University of KwaZulu-Natal, Durban, South Africa, 8 Department of Mental Health Nursing, Kamuzu University of Health Sciences, Blantyre, Malawi, 9 Centre for Reproductive Health, Kamuzu University of Health Sciences, Blantyre, Malawi, 10 Department of Psychiatry, University of North Carolina at Chapel Hill School of Medicine, Chapel Hill, North Carolina, United States of America

* mastockt@email.unc.edu

## Abstract

Psychotic disorders are highly stigmatized across the globe, negatively impacting people with psychosis and their families. However, little is known about stigma faced by people with psychosis in sub-Saharan Africa. We developed semi-structured qualitative guides based in a constructivist epistemology and formative research methodologies and conducted 36 in-depth interviews (IDIs) and two focus-group discussions (FGDs) with 12 people with lived experience (PWLE) with psychosis; 12 caregivers of PWLE; six traditional healers; six medical providers; six community leaders (1 FGD); and six religious leaders (1 FGD) in Blantyre, Malawi. We drew from the Health Stigma and Discrimination Framework to delineate the stigmatization process. Participants described key drivers of stigma as lack of awareness, prejudice, stereotypes, and fear. Manifestations included experienced, anticipated, witnessed, perceived, internalized and secondary stigma in the form of insults, gossip, abuse, physical violence, restraints, social exclusion, and employment-based discrimination from family and community. With respect to negative outcomes and health and social impacts, stigma impacted quality of care, resilience, mental health, morbidity, social inclusion and quality of life. In Malawi, stigma is pervasive challenge for PWLE, with severe implication for their health and social wellbeing. In partnership with PWLE, investment into the integration of evidence-based stigma reduction activities into existing psychosis management programs is warranted.

**Data availability statement:** Our qualitative data set is not available for sharing publicly because we do not have ethical approval to share beyond our immediate research team. The Research Ethics Boards who have imposed this restriction are: University of North Carolina at Chapel Hill (contact: irb_questions@unc.edu); the National Health Science Research Committee (contact: directorgeneral@ncst.mw).

**Funding:** This study was funded by the National Institute of Mental Health (R34MH131234 to KK and BW; K01MH130226 to MS; and D43TW011794 to PN, JN, and MN). This content is solely the responsibility of the authors and does not necessarily represent the official views of the National Institutes of Health. The funders had no role in study design, data collection and analysis, decision to publish, or preparation of the manuscript.

**Competing interests:** The authors have declared that no competing interests exist.

## Introduction

Psychosis is one of the leading causes of disability worldwide, contributing to the burden of disease in low- and middle-income countries (LMIC) [1,2]. Prevalence estimates of psychotic disorders in sub-Saharan African countries range from 0.3% to 5% [3]. While, data on the prevalence or incidence of psychosis in the region is limited [4], patients commonly present with psychotic illness at Malawian psychiatric hospitals. One study found that over a 12-month period, 30% of patients admitted to the psychiatry department at a district hospital had been diagnosed with a psychotic disorder [5]. In settings such as Malawi, there is under-allocation of resources to mental health services, a shortage of trained psychiatric providers, and high usage of alternative healthcare systems [6–9]. People with psychosis may not receive critical care and support, and thus have worse morbidity and mortality compared to the general public [4,10–12].

Health inequalities faced by individuals with psychosis may be exacerbated by stigma [13]. Psychotic disorders are highly stigmatized across the globe, negatively impacting those affected and their families. Those with psychotic disorders are often discredited and labeled as "lesser" due to their behavioral symptoms [14,15]. This reduced social status is a result of the tendency to attribute behaviors to one's internal characteristics rather than situational circumstances—a phenomenon that is common in the stigmatization process for mental disorders [14–17]. In Malawi, people often make this attribution error, attributing behavioral symptoms of psychosis to perceived faults such as the utilization of witchcraft or substance use. Stigma occurs within a context of unequal power, conceptualized as a process of 1) distinguishing and labeling differences (*person who is mentally ill*); 2) associating negative attributes (*dangerous, unpredictable, etc.*); and 3) separation (*physical and social isolation*), culminating in 4) status loss and discrimination (*abuse, disrespectful treatment, denied opportunities*) [15,17–19]. The Health Stigma and Discrimination Framework posits that health-related stigma manifests across socio-ecological levels and can detrimentally impact people's health and well-being [20]. This framework delineates the stigmatization process in the context of health, exploring the drivers and facilitators of health-related stigma, the manifestations of this stigma, and the impact of stigma on the individual-, interpersonal-, organizational-, community-, and public policy-levels. Recognizing the cultural and contextual nuances, the Health Stigma and Discrimination Framework provides a model for the universal process of stigmatizing health conditions, making it an ideal framework for this study [20]. While there are ubiquitous aspects of the stigmatization process across health conditions,[19] stigma related to psychosis and its impact on wellbeing remain culturally and contextually specific.

In Malawi, recent studies on mental disorders indicate that the prevalence of the stigmatization of mental health conditions is high as demonstrated by elevated rates of internalized stigma and mistreatment and discrimination by families, communities, and healthcare workers [8,21–23]. In Africa, psychosis stigma can result in decreased self-esteem, development of other mental disorders such as depression,

delayed access to care, disrupted adherence to treatment, diminished job and relationship opportunities, and increased psychiatric symptoms [8,23–30]. For example, internalized and perceived stigma were associated with reduced medication adherence among those with schizophrenia in Ethiopia [10,26,31]. Stigma acts as a barrier to rehabilitation and is associated with poor recovery for people with severe mental disorders [23,24,32,33]. Family members of people with psychosis are also subject to stigmatization, exacerbating the burden of caregiving [34–36]. Understanding stigma is fundamental to optimizing treatment, improving health outcomes, and protecting the human rights of people with psychosis, particularly in resource-limited settings. Yet, little is known about stigma faced by people with psychosis in sub-Saharan Africa.

In this manuscript, we describe stigma and discrimination faced by people with psychosis in Malawi using the Health Stigma and Discrimination Framework to delineate the stigmatization process including drivers, manifestations, outcomes, and health and social impacts of stigma [20].

## Methods

### Ethics statement

The University of North Carolina IRB and the Malawian National Health Sciences Research Committee approved the study protocol. All study participants provided written informed consent and received the equivalent of 10 USD for participation.

### Setting

The study was conducted at Queen Elizabeth Central Hospital (QECH) in Blantyre, Malawi. One of four central hospitals in Malawi, QECH provides tertiary specialized care and operates the largest outpatient mental health clinic in the country. QECH serves the peri-urban population of Blantyre District and surrounding districts.

The psychiatric outpatient clinic at QECH provides care to an average of 600 patients each month. Approximately a quarter of these patients have received a severe mental disorder diagnosis. Most outpatients with a psychosis diagnosis are followed in the post-acute period, meaning after inpatient hospitalization and discharge. The clinic is managed by two psychiatrists, five psychiatric trainees (medical officers) and five mental health nurses.

### Study population and recruitment

Eligibility criteria for the study were defined as being ≥18 years of age, residing or working in Blantyre District, and belonging to one of the following groups: (1) people with lived experience (PWLE) with psychosis (confirmed by clinician diagnosis) receiving outpatient psychiatric care; (2) caregivers of PWLE; (3) community leaders; (4) religious leaders; (5) traditional healers; and (6) mental health providers who provide outpatient psychosis services.

PWLE and their caregivers were recruited from QECH. The psychiatric clinic staff recruited PWLE during their clinic visit by screening PWLE with the Clinical Global Impression-Severity Scale (CGI-S) [37]. The CGI-S scores were verified by a psychiatrist. The PWLE sample was stratified between individuals with current elevated psychotic symptoms, defined as a CGI-S score ≥3, and those with minimal or subthreshold psychotic symptoms, defined as a CGI-S score ≤2. Caregivers were recruited by asking PWLE for permission to speak with their caregiver.

The study coordinator worked with two senior traditional chiefs to recruit community leaders in Blantyre District and worked with an interreligious Christian association called Act Alliance to recruit the religious leaders. To recruit traditional healers, study staff coordinated with the president of the traditional healers' association of Malawi (MTHOU) to identify one traditional healer from each of the six traditional administrative areas in Blantyre.

The study coordinator identified a convenient sample of six medical personnel who were present, available and willing to complete interviews.

PLOS Mental Health

### Data collection tools and process

The research team developed semi-structured qualitative guides based in a constructivist epistemology and formative research methodologies [38], with the overarching goal of gathering data to adapt a community-based rehabilitation intervention. Congruent with constructivists principles, the guides used open-ended prompts to elicit how participants consider, interpret and make meaning out of their experience with psychosis, with a particular focus on developing a deeper understanding of how individuals in Malawi conceptualize their illness, rehabilitation and recovery. In alignment with the study objective, the guides were organized as follows: 1) understanding of and experiences with psychosis; 2) care seeking and pathways to care; 3) perceptions of and experiences with stigma; 4) treatment, rehabilitation and goal setting; and 5) preferences for community-based rehabilitation care.

Trained interviewers piloted the guides with PWLE and caregivers who were not included in the study sample. Ultimately, in-depth interviews (IDIs) were conducted with PWLE, their caregivers (CG), traditional healers (TH) and mental health providers (MP) and focus group discussions (FGDs) were conducted with community leaders (CL) and religious leaders (RL).

Data were collected in March and April 2023 at the QECH outpatient clinic. Trained interviewers conducted the IDIs and FGDs in Chichewa, which lasted approximately one hour. All IDIs and FGDs were audiotaped, transcribed, and translated into English. The research team reviewed transcripts as they became available, provided feedback to the interviewers and revised the interview guides throughout the data collection process.

### Analysis & analytic framework

We used Dedoose [39] to process the data. After reading all transcripts, GS drafted a thematic codebook that captured emerging themes. Such themes included education and employment discrimination; mistreatment by community, family, or health workers; PWLE violence toward others or disruptive behaviors; disclosure to family members and community; stigmatizing attitudes; and coping strategies. The coders (GS, AM, HT, JD, and JK) read and coded a subset of the same transcripts (4 of the 38) independently to ensure consistency in coding. Coding was treated as an iterative process wherein the coders met multiple times to discuss and reach consensus on the addition, definition, and appropriate use of the codes. After coding was complete, MS and JK reviewed all coded data related to stigma and discrimination, by participant group. MS and JK then conducted thematic analysis [40] guided by the Health Stigma and Discrimination Framework [20].

The Health Stigma and Discrimination Framework describes the stigmatization process in the context of health [20]. The framework encompasses the individual, interpersonal, and structural levels of health stigma, dividing the process into a series of domains that include drivers, manifestations, outcomes, and health and social impacts. Drivers are the underlying factors that cause stigma, such as societal norms and cultural beliefs. Different types of manifestations of stigma may include: experienced stigma or discrimination (e.g., verbal abuse, gossip), anticipated stigma (i.e., expectation or fear of discrimination), perceived stigma (i.e., perceptions about how stigmatized groups are treated in a given context), internalized stigma (i.e., acceptance of negative societal attitudes associated with one's stigmatized status as true and just), and secondary stigma (i.e., stigma felt by those associated with the stigmatized group such as family or healthcare providers). Stigma manifestations result in negative outcomes that culminate in larger negative health and social impacts on mental health, social inclusion, quality of life, and mortality. The framework was used to consider, organize and ultimately present themes related to drivers, manifestations, outcomes, and health and social impacts of psychosis stigma and discrimination.

## Results

### Participant characteristics

Participants included PWLE (n = 12), caregivers (n = 12), traditional healers (n = 6), medical personnel (n = 6), community leaders (n = 6), and religious leaders (n = 6). (Table 1). The majority of PWLE participants were men (n = 9). Seven PWLE were diagnosed with schizophrenia, four with schizoaffective disorder and one with an unclassified psychotic disorder. The majority (9/12) had either minimal or subthreshold current psychotic symptoms (CGI-S≤2). The average time since

first onset of psychosis symptoms was 11.6 years (Range 7–18). Caregivers were most often parents or siblings of PWLE. Most caregivers were women (n = 8). Educational achievement among both PWLE and caregivers was low with lower achievement among caregivers. There was an equal number of men and women traditional healers with an average age of 61 and generally low educational achievement. Among medical personnel, three medical officers and three nurses participated; two medical officers were men, while all three nurses were women. The community leader FGD was primarily composed of men (n = 5) with low educational achievement. The religious leader FGD had an equal number of men and women participants with high educational achievement; all were members of a Christian denomination.

## Stigma and Discrimination faced by people with psychosis in Malawi

Drawing from the Health Stigma and Discrimination Framework, we present the themes related to key *drivers* of stigma (lack of awareness, prejudice, stereotypes, and fear); *manifestations* of different types of stigma (experienced, anticipated, witnessed, perceived, internalized and secondary); the *negative outcomes* of stigma on access to healthcare and reliance; and ultimately the *negative health and social impact* on mental health, social inclusion, quality of life, and morbidity (Fig 1).

**Adapted from the Health Stigma and Discrimination Framework.** **Drivers**: Key drivers of stigma were rooted in interlocking lack of knowledge, prejudice, negative stereotypes, and fear.

**Lack of knowledge.** All participant groups broadly acknowledged low biomedical mental health knowledge among the public and a severe lack of knowledge about psychosis as a clinical condition or about the prospect of recovering from psychosis. This lack of knowledge at the community level was explicitly recognized as a key driver of stigma. For example, participants reported that people with psychosis may be mistreated by the community simply because community members do not understand mental illnesses and/or why people with psychosis behave or act differently than others. As heard from one MP, *"I think they do this* [discriminate] *because of lack of knowledge. They do that because they don't know what mental illness is"* (MP-03). This MP links lack of knowledge to stigmatizing behaviors. A caregiver also explained how community members *"did not know that his* [their son's] *abnormal behavior was due to a mental illness. So he went to this other place and he started misbehaving and they hurt him. He has a scar on his head. They hit him without knowing that they are hitting someone with a mental illness"* (FM-01). Further, participants reported community-wide beliefs that people with psychosis could not recover or become stable. *"My own relatives did not know that due to my condition, I can be in this stable state, they thought that I was going to die due to this mental illness"* (PW-10). In an absence of knowledge, prevailing perceptions of psychosis were dominated by inaccurate negative stereotypes and prejudiced beliefs.

**Table 1. Participant Characteristics.**

| N or Mean (Range) | PWLE | Caregivers | Traditional Healers | Medical Personnel | Community Leaders | Religious Leaders |
|---|---|---|---|---|---|---|
| Total | 12 | 12 | 6 | 6 | 6 | 6 |
| Gender | | | | | | |
| Women | 3 | 8 | 3 | 4 | 1 | 3 |
| Men | 9 | 4 | 3 | 2 | 5 | 3 |
| Average Age | 37 (23-49) | 45 (24-58) | 61 (42-82) | 37 (27-56) | 62 (54-69) | 52 (48-57) |
| Education | | | | | | |
| None | 0 | 1 | 0 | 0 | 0 | 0 |
| Primary | 2 | 4 | 4 | 0 | 4 | 0 |
| Secondary | 8 | 5 | 2 | 0 | 1 | 1 |
| Tertiary | 2 | 2 | 0 | 6 | 1 | 5 |

## DRIVERS

**Lack of Knowledge:** low mental health literacy, lack of knowledge about psychosis and recovery

**Prejudice:** beliefs that psychosis is caused by substance use, witchcraft and/or possession

**Negative Labels & Stereotypes:** labels of dangerous, violent, aggressive, unpredictable, incompetent, useless, not normal, dirty, less than human

**Fear:** fear of violence/unpredictability, fear of transmission of psychosis

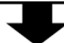

## MANIFESTATIONS

**Experienced Stigma:** insults, gossip, name-calling, abuse and physical violence, restraining, avoidance, social exclusion, employment-based discrimination

**Witnessed/Heard Stigma:** verbal abuse, physical and sexual violence, restraining, social exclusion, education-based discrimination, abuse at the hands of family members

**Anticipated Stigma:** expectation of stigma if psychosis diagnosis was disclosed, maintaining secrecy to avoid stigma from community and employers

**Perceived Stigma:** perception of psychosis stigma negatively impacting ability to form and maintain platonic and romantic relationships

**Internalized Stigma:** shame, low self-esteem, negative self-perception as "mad"

**Secondary Stigma:** family members mocked, insulted, blamed for causing psychosis through neglect or witchcraft, diminished marriage prospects

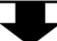

## NEGATIVE OUTCOMES

**Access to Quality Healthcare:** insults, physical violence, inappropriate restraining at health facilities, stigmatizing attitudes among general staff

**Resilience:** coping by spending time with friends and family, praying, exercising, listening to music, watching television, performing chores/jobs, acceptance of psychosis

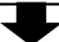

## NEGATIVE HEALTH & SOCIAL IMPACTS

**Mental Health:** worsened mental health among people with psychosis and family members

**Social Inclusion:** exclusion from community social activities, events, or general interaction

**Quality of Life:** negatively impacted relationships, social belonging, social status, ability to generate income

**Morbidity:** stigma believed to worsen health outcomes

**Fig 1. Psychosis Stigma and Discrimination Framework.**

**Prejudice (i.e., negative evaluation of people with psychosis).** Participants from all groups reported negative beliefs about the etiology of psychosis at the community level; namely, in the community psychosis is believed to be caused by substance use, witchcraft and/or possession.

Many participants reported that substance use, particularly marijuana or "Chamba," is a cause of psychosis. Witchcraft or misuse of traditional medicine were also believed to cause psychosis, particularly when someone, or their relatives, tried to use witchcraft for nefarious purposes or personal gain. Misusing witchcraft or traditional medicine to "get rich quick" or "get ahead in life" was often mentioned, as described by a traditional healer: *"People do have these beliefs saying that maybe in order to get rich quickly the person living with madness was using traditional medicine (kugwila zitsamba) and as a result the person got mad"* (TH-05). Others believed psychosis was caused by demonic or evil spirit possession, associated with having sinned or with satanism. For example, an individual with psychosis explained that some people *"think that you have been bewitched or a spell has been placed on you and that is why you are having that mental disorder. Others also believe that it is the Satanists that are part of a satanic cult that are impacting your mental wellbeing negatively"* (PW-02). In these statements, the participants demonstrate prevalent prejudiced attitudes around alcohol and drug use, misuse of witchcraft/traditional medicine, and sin. The implicit assumption that the individual is responsible for their condition and can be "blamed" for their bad behavior.

**Negative labels & stereotypes (i.e., characteristics associated with people with psychosis).** People with psychosis reported being called: "*openga/wapenga*" (crazy/mad person), "*wamisala*" (mad one), "*ozungulila mutu*" (someone whose head does not work), "*wa Zomba*" (person belongs in Zomba Mental Hospital), "*wa mental*" (person with mental health challenges), among other names. One participant even reported being referred to by community members only as a "*misala*/mad person" instead of their actual name. In this way, the labeling of people with psychosis demonstrates the lack of neutral or polite terms in Chichewa to describe psychosis, with nearly all terms used carrying a negative or derogatory connotation.

Participants from all groups associated negative stereotypes to individuals with psychosis, noting such individuals would be further labeled as short-tempered, dangerous, violent, aggressive, unpredictable, untrustworthy, incompetent, useless, "good for nothing," not normal, disorganized, dirty, and unclean. Among these stereotypes, the belief that people with psychosis are dangerous and violent was most prevalent. An MP described how community members generally believed "*mad people are very violent and don't ever interact with them because you get beat up or something … that belief is very rampant.*" (MP-06).

With respect to stereotypes around the capabilities of individuals with psychosis, generally people with psychosis were thought to be unable to work, contribute to their families or communities or fulfill societal roles such as getting married and having a family. As described by one individual with psychosis, "*people in our communities consider us as useless because they say these people* [with psychosis] *are sick so they cannot do anything. For example, they cannot work because they are diagnosed with psychosis. Also, we cannot participate in the development activities in the communities. They discriminate us so that affects us heavily*" (PW-01). In this scenario, the stereotypes around individuals' competence and ability to contribute are believed to result in discrimination in the form of exclusion from economic development opportunities.

Stereotypes around one's inability to contribute or recover erode other's hope and trust in people with psychosis. One community leader described, *"Most of the community members lose hope in the person with psychosis and they discriminate against him because they believe that he is mad and he is no longer like them since he is mentally unwell. Yeah, even after the patient recovers from psychosis people still don't trust him as they used to before; …They don't involve him in the community activities citing that he is a mad person."* (CL-06). This statement demonstrates how negative stereotypes around productivity, erosion of trust, and a lack of understanding around the potential for recovery drive discrimination.

The sum of these negative stereotypes allowed people with psychosis to be assumed to be less than human, as described by two traditional healers*: "the person living with madness has already lost their humanity"* and *"it's like the*

*person is no longer a human being but an animal"* (TH-05, TH-06). These descriptions demonstrate how individuals with psychosis can be considered to have lost their right to human dignity.

**Fear.** Fear, in part stemming from lack of knowledge, prejudice and negative stereotypes, also drove stigmatizing behavior. Fear of the individuals themselves due to concerns of violence and unpredictably was commonly described as causing discriminatory behavior, particularly avoidance and social exclusion. One MP described, *"people fear people who are showing disorganized behaviors because they just fear that they are capable of doing anything, yeah, so they* [patients with psychosis] *are people who are feared by the community"* (MP-04). A caregiver also described, *"people believe that people with psychosis are dangerous and unpredictable because they just do things unexpectedly at any time. With that in mind, people consider someone with psychosis as a dangerous person and in fact, when a person with psychosis is passing by people always try to stay away from him"* (CG-01). Participants also reported fear of transmission of psychosis resulting in avoidance and social exclusion. As heard from a traditional healer, *"[community members] start running when they see the person* [with psychosis]*. They say that the person can infect them with psychosis"* (TH-04). A caregiver concurred, saying community members *"keep their distance from people that are living with this mental disorder… They think that they can also become mad by hanging around"* (CG-09). In this manner, unwarranted fear of psychosis 'transmission', in addition to fear of violence and unpredictability, drove community members to unjustly treat people with psychosis.

## Manifestations

**Experienced stigma.** Participants reported that people with psychosis experienced stigma and discrimination in the form of insults, gossip, name-calling, abuse and physical violence, being restrained or tied up, avoidance and social exclusion, and employment-based discrimination.

People with psychosis were often subjected to ridicule, stigmatizing language and unkind behavior including mocking, name-calling, and being laughed at. As one individual described, *"all they do is insult you that you are mad. Everyone will just be saying there goes the mad person when you are walking in the road. So such things are very painful indeed"* (PW-06). Many participants described similar derogatory experiences. While verbal abuse mainly was perpetrated by community members, some participants reported insults and mockery by family as well. *"Of course, there are some family members who still discriminate against me by undervaluing me, for example, saying that since I have psychosis, it means there is nothing I can do to help in the family"* (PW-01).

People with psychosis are particularly vulnerable to physical violence. Participants reported people with psychosis being physically assaulted by both community and family members, though instances of abuse perpetrated by community members were more common. Some participants reported an increased vulnerability of women with psychosis to sexual assault and rape by community members; *"When it is the females* [with psychosis]*, other men are able to rape them"* (PW-06). Some participants believed people with psychosis might face physical violence in retaliation for undesirable, inappropriate, or destructive behavior. However, more commonly people with psychosis were beaten "for no reason." As one caregiver described, *"These were people that were just being cruel to him, there were beating him up for no reason at all, maybe it was because they knew that he could not fight back. They still beat him up even though he did not provoke them… they were doing so because they did not know him and they did not understand his mental condition"* (CG-04). In this instance, a vulnerable individual, unable to defend themselves, is subject to stigma that manifests as unprompted violence.

Another common form of abuse was restraining or tying up of individuals with psychosis. Individuals with psychosis reported being bound with rope when "behaving inappropriately" or "being destructive." Restraining was used by family and community members to prevent people with psychosis from harming themselves and others, running away, or damaging property. The practice was often cited as a method to facilitate visits to the hospital in times of crisis. One individual recounted, *"I cannot tell what exactly are the symptoms that I experienced but maybe I just realized that people have*

*grabbed and tied me and are taking me to the hospital. In fact, they don't say 'we are taking you to the hospital,' they just tie me and tell me 'you need help.' I could tell them 'no, I don't need help I am alright. My family members just take me to the hospital anyways.'"* (PW-05). In this statement, the participant describes how being tied up removed their agency as they were physically bound and taken to the hospital against their will.

Employment-based discrimination was experienced by some participants with psychosis. Participants reported being denied job opportunities and requests for capital to start a business due to their mental illness. Employment-based discrimination was generally understood to arise from stereotypes surrounding the inability to work. An individual with psychosis described how employers, "*when they know that you are mentally ill, they don't recruit you saying that 'we need people who are mentally well to do the job'*" (PW-01). Similarly, another individual with psychosis explained, "*Since the person does not have an opportunity to do any work, if the person goes somewhere and tries to do some piece work, the people will not let that person work*" (PW-06). In this way, even those trying to work outside of the formal employment sector face discrimination-driven barriers to generating income.

**Witnessed/Heard stigma.** CG, MP, TH, CL, and RL participants reported hearing about or witnessing manifestations of discrimination, corroborating the experiences of people with psychosis detailed above. Additionally, participants from these groups reported other stigmatizing and discriminatory behavior, including education-based discrimination and abuse at the hands of family members.

One caregiver described the education-based discrimination experienced by their son with psychosis. "*Even in the classroom the teacher … makes him sit outside to write his exams while his classmates are inside writing the same exams. He is forced to look inside through the window in order to see the exams being written. All this because they say that he is mad and he can end up disturbing his classmates*" (CG-06). Additionally, this participant recounted how their son had been whipped by educators. In this instance, negative stereotypes surrounding the disruptive behavior of people with psychosis compromise the quality of education.

In addition to accounts of people with psychosis being tied up and beaten by family members, multiple MP, TH, CL, and RL participants recounted hearing of family members not giving individuals with psychosis food. A TH described, "*Some people with psychosis are not given food by their caregivers*" (TH-04). An RL also reported, "*others are chained and others are beaten and maybe others could be denied food.*" (R-06). One CL explained that this treatment can lead to people with psychosis being left alone to fend for themselves by looking through trash or begging for food, increasing vulnerability and reinforcing negative stereotypes.

**Anticipated stigma.** Participants feared stigma following disclosure of one's psychosis. In fact, many individuals and caregivers kept the diagnosis secret to avoid stigma and discrimination from the community and current (or future) employers.

Many individuals with psychosis and caregivers cited anticipated stigma as a reason not to disclose their diagnosis to the community. When asked why they did not openly share their diagnosis, one individual responded, "*I just have a feeling that if everyone was to know about me having a mental disorder then the people will be stigmatizing me*" (PW-12). Another individual concurred, *"I have started hiding my condition because if others know, there are other people that change the way they interact with me"* (PW-02). A caregiver also described hiding their relative's psychotic disorder to protect them from stigma: "*He will be stigmatized. They* [community members] *will not be chatting with him the way they used to once they find out about his mental condition*" (CG-01). In addition to talking with people with psychosis differently, participants expected insults, mockery, rumors, and "pointing fingers" from the community as well as employment-based discrimination. These statements demonstrate anticipated negative changes in community attitudes and behaviors toward people with psychosis following disclosure.

**Perceived stigma.** Participants commonly reported perceived stigma that impacted both platonic and romantic relationships. With platonic relationships, participants generally believed it was difficult for people with psychosis to form and maintain friendships due to stigma. One individual with psychosis described, "*I do not think that people would*

*want to be close friends with someone that is living with this mental disorder*" (PW-12). With romantic relationships, participants widely acknowledged that people with psychosis had diminished marriage prospects. A caregiver described how community members conclude marriage is impossible: *"They might think that the person with the mental illness will not take care of his family as a married person is supposed to. And when they have children, they will not be able to take care of them"* (CG-01). A traditional healer also said that "*there isn't anyone else that could accept them* [people with psychosis]" for marriage "*even after recovering, because the people still consider the person dangerous saying that, maybe someday their psychosis could return and they would hurt them. And that's why women don't marry men with psychosis.*" (TH-02). The sentiment that people with psychosis are unable to be married was fueled by negative stereotypes surrounding their ability to maintain relationships, support a family, or recover as well as by fears of violent behavior. Fear was a more disabling factor in marriage prospects for men compared to women.

**Internalized stigma.**  Participants with psychosis internalized stigma in the form of shame, low self-esteem, and considering themselves as a "mad person' rather than a "normal person". Internalized stigma was commonly described as stemming from stigma from the community. As one individual described, "*You feel like people are stigmatizing you. When you are walking, the people seem to be minding your business because you see the people talking about you. As a result, you start to stigmatize yourself because you feel like the people do not love you or do not care about your presence*" (PW-06). This statement highlights the process through which social alienation and gossip lead to the internalization of stigma. Another individual noted how negative self-perceptions developed: "*The challenge is the thought that you were born a normal person and then later in life you become psychotic, and people start calling you a mad person, so you just consider yourself different from all your friends*" (PW-05). In this way, individuals begin to view themselves as no longer "normal" or different due to continued labeling from community members.

**Secondary stigma.**  Most participants did not report experiencing, witnessing, or hearing about stigma directed at family members of people with psychosis. However, some members from each participant group did acknowledge that relatives of people with psychosis experience severe stigma. More commonly, family members anticipated stigma and feared the consequences of disclosing their relative's diagnosis.

Some caregivers reported being blamed for causing their relative's psychosis through neglect or witchcraft: "*they* [community members] *say that you wanted to become rich by making someone have mental illness … Whenever there is some sort of development at your household, then they believe that the development has been possible because of the mental illness of the child*" (CG-03). In this instance, the community looks down on the family's prosperity, as the family's success is attributed to inducing their relative's psychosis. Family members may also be blamed for the actions of their relative. For example, "*As family members you are also mistreated, you are ridiculed from surrounding people, especially those that your family member with psychosis has damaged their properties or their goods*" (R-01).

TH, CL, RL and particularly the MP participants described how the community would mock and insult family members of people with psychosis. One MP gave the example, "*someone has a sister with psychosis. People can use this sibling who has psychosis to mock the other normal sibling to say 'your sister is mad (wamisala) so you will also get mad some time'*" (MP-04). A community leader described another example: "*if their father or mother has psychosis, children are discriminated and despised by people because their mother or father is mad. If the children make any mistake, people mock them that they have inherited their parent's psychosis*" (CL-03). A MP further described how mistreatment stems from how "*people also think that members of the family also have problems in regards to their thinking capacity. People tend to … transfer the perceptions that they have towards someone suffering from psychosis to his or her relatives.*" (MP-03). In these examples, the relatives of people with psychosis experience verbal abuse, which often centers around beliefs about the relatives' intelligence or the possibility of also developing psychosis.

Some MPs discussed how having a relative with psychosis can diminish family members' marriage prospects. One MP described how some community members planning to marry someone feel they "*have to screen their background for psychosis*" due to the belief that "*if the person has a relative in their family who has psychosis, then they might also pass*

*those traits of psychosis to the children they might have*" (MP-04). The fear of psychosis being passed genetically was cited by multiple MPs as negatively impacting the marriage prospects of relatives of individuals with psychosis.

Further, participants reported how family members of people with psychosis anticipated stigma. A community leader explained, "*they* [the family] *fear being discriminated and despised by the community members*" (CL-05). Such fear of stigmatizing behavior resulted in feelings of shame and embarrassment surrounding disclosure. As one traditional healer noted, parents of a child with psychosis "*feel ashamed saying that, when they say that the child has psychosis, then it means that they neglected the child to smoke marijuana, and that's what most parents think*" (TH-02). An MP also mentioned that family members may not want to disclose their relative's mental illness "*because they don't want people to know that they are related to a mad person who is perceived as incompetent and who is less than a human. It also might have negative implications on the family because people might think that the family is bewitched and things like that*" (MP-06). As these statements demonstrate, family members of individuals with psychosis often fear secondary stigma following disclosure of their relative's mental illness.

### Negative outcomes

Psychosis stigma manifestations had a significant influence on access to quality healthcare for people with psychosis, particularly for non-psychiatric care, as well as implications for resilience in the face of stigma.

**Access to quality health care.** Most individuals with psychosis did not report experiences of mistreatment by healthcare providers. However, one participant did report being insulted and slapped by a healthcare provider, saying "*when I was admitted at this other hospital, a healthcare worker slapped me...she just said 'you are rude' then slapped me*" (PW-03). Other PWLE reported what they believed to be inappropriate restraining.

Some medical practitioners discussed how psychosis stigma, particularly in communities, hindered access to treatment for people with psychosis and the quality of care received. For example, multiple MPs discussed how non-psychiatric specialists (e.g., those not involved in mental healthcare) do not provide adequate general care for people with psychosis. One MP described, "*healthcare workers sometimes do not want to treat these people* [who have psychosis] *or provide medical care. People who had psychotic symptoms or are showing symptoms of psychosis are more likely to get poor services from the healthcare workers in general*" (MP-05). Another MP described, "*at times there is that level of stigma towards people with psychosis. … For example, the Nurses would not be checking up on them, not giving them their timely medications in the general wards because they would say 'these people with mental illness, they belong to the psychiatric unit.' Yeah, so with that attitude, you will find out that they* [patients with psychosis] *will not receive the proper care*" (MP-04). These statements suggest that stigmatizing attitudes among general staff may hamper patients' ability to access non-mental healthcare and decrease the quality of care provided, particularly outside of the psychiatric ward.

**Resilience.** Participants with psychosis described the various mechanisms used to cope and remain resilient in the face of stigma. These methods include spending time with friends and family, cited as sources of social support, praying, engaging in activities such as exercising, listening to music, watching television and movies, performing household chores and part time jobs, and spending time on the Internet. Acceptance of their mental disorder was an important resiliency factor. One participant described, "*I choose to leave a peaceful life by doing things that are good for me like going to church … Sometimes I do physical exercises at home, things like press-ups, so that my body can be having energy… I do not spend my time paying attention to what people might be saying about me. I make sure that such things do not get to me, so I avoid getting worried because of the actions of other people.*" (PW-06). This individual remains resilient by engaging in activities that are important to them that simultaneously serve as a distraction from negative interactions with others. Many other participants described similar strategies of avoiding conflict and trying not to pay attention to what others say or do regarding their mental condition.

Other participants focused on accepting their mental condition and working toward recovery. "*Even when they* [community members] *know and even if they laugh at you, what can you do about it? You just need to keep doing your work as*

*well as take your medication appropriately. That is the only thing to focus on*" (PW-10). In accepting their psychosis and focusing on treatment adherence and recovery, this individual was able to stay resilient in the face of stigma.

### Negative health and social impacts

Regarding health and social impacts, psychosis stigma manifestations and the associated healthcare and resiliency outcomes further impacted mental health, morbidity, social inclusion, and quality of life.

**Mental health.** Psychosis stigma affected participants' mental health. As one individual described, "*Sometimes I was being verbally abused by having people call me wamisala (mad person) or ameneyo wa mental (person with mental issues). As a result, I was feeling pain in my heart wondering why these people are insulting me in that way.*" (PW-02) In describing "*pain in my heart,*" this participant uses a colloquialism often heard from patients describing depressive symptoms in Malawi.[37] A caregiver described how overhearing gossip about "his ruined future" caused their child to become depressed. "*You know sometimes people are able to talk … saying that, 'the child is intelligent and he would have been a boss.' And when* [my child] *comes back home, he comes depressed saying that people were talking about him*" (CG-03). A provider even noted how "*because of the discrimination and isolation being faced, then they are able to get into depression and others even commit suicide*" (MP-02). A community leader similarly described the toll verbal abuse can take on family members: "*The family members are affected because they are stigmatized by the community members … So, they are mentally affected*" (CL-04). In this way, stigma negatively impacts the mental health of both people with psychosis and their families.

**Morbidity.** The above descriptions of the adverse impact of stigma on health and wellbeing imply that individuals connect stigma with worsening health outcomes for people with psychosis. While not explicitly stated by many participants, one caregiver described how stigma following disclosure of one's psychosis diagnosis would "*cause psychosis to get worse, therefore it's better just to stay and don't disclose*" (CG-05). As such, stigma is believed to worsen the trajectory of individuals with psychosis.

**Social inclusion.** People with psychosis reported various forms of social exclusion due to their mental illness, particularly from community activities, social events, and generally interacting with community members. As described by one PWLE, "*we are discriminated against. For example, when there are programs being implemented in our communities we are not involved because people say we are mad people.*" (PW-01). People with psychosis are excluded from programs that often aid in economic development, restricting their opportunities for economic mobility. Others described exclusions from both social events, such as weddings and funerals, as well as daily interactions. Many participants described the community's general avoidance of people with psychosis, including running away from them, refusing to befriend them, or simply ignoring them in public. One individual detailed how some friends from school now "*say that I am a mad person and they cannot be associating with me. Even when I greet them, they do not respond to my greetings. They just walk on by like I am not even there just because they know that I suffer from a mental disorder*" (PW-02). Due to their mental illness, people with psychosis can be excluded and disconnected from social settings and denied important social relationships and a sense of belonging or value in their community.

**Quality of life.** Many participants with psychosis reported experiencing decreased quality of life due to psychosis stigma. As described in the above sections, stigma negatively impacted participants' relationships, social belonging, social status, and ability to make money. Multiple participants spoke about how psychosis "ruined" their reputations such that they had no friends, no dating or marriage prospects, and limited opportunities to attain wealth.

## Discussion

In this setting, inter-related and reinforcing drivers of stigma included limited knowledge about mental health and recovery, beliefs and blame around the etiology of psychosis, negative stereotypes, and fear. These ideas persist in Malawi, as shown by decade-old findings documenting similar beliefs around substance abuse, spiritual possession, and God's

punishment as causing psychosis [22]. Other studies have also identified lack of knowledge as reinforcing prejudice, stereotyping and fear [22,41–45]. In particular, in settings where substance use and witchcraft are believed to cause psychosis or where mental illness is considered more of a character failing than a disease, community members blame or hold individuals (and/or their family members) responsible for their condition. This blame, combined with underlying fear, can form the basis for justifying stigma and discrimination [22,41,43].

In identifying the key drivers of stigma (lack of knowledge, stereotypes, etc.), there is an opportunity to reduce stigma by addressing the root causes of discriminatory behavior [30,46]. While few interventions to address mental illness stigma have been developed in LMICs [47], contact-based strategies where PWLE interact with others, participate in trainings and/or share testimonials can build empathy, improve attitudes, and challenging stereotypes and prejudice [48–51]. Such interventions could be bolstered by efforts to correct inaccurate knowledge about psychosis and the potential for recovery with treatment, which could prevent blame, challenge inaccurate stereotypes, and decrease fear [30]. In this setting, the family and community are both a key source of material and social support, as well as key perpetuators of stigma. As such, community-based rehabilitation programs for families that focus on improving attitudes and knowledge while also engaging community and religious leaders, may have important implications for stigma reduction [52]. Ultimately, multi-pronged stigma reduction approaches that target the key drivers of stigma and are conducted in partnership with PWLE and their families are needed in low-resource settings such as Malawi.

Findings from this study elucidate the many manifestations of stigma faced by people with psychosis and the negative impact of stigma on health and social outcomes. Other regional studies have similarly documented negative attitudes towards people with mental illness, general mistreatment (mocking, abuse, etc.), different forms of social exclusion of people with psychosis, and self-stigma [41,43,53,54]. Participants in this study described how experienced, anticipated and perceived stigma from society could lead to the internalization of stigma, a common occurrence among people with mental health disorders in low-resource settings [55]. Stigma, in particular internalized stigma, has been linked to anxiety and depression, worse engagement in mental healthcare, more severe psychiatric symptoms, poor recovery, and a lower quality of life as it severely impacts the ability to perform daily activities and carry out roles that matter to the individual [44,55–60]. These negative health outcomes are further complicated by the cultural convention to seek traditional medical care before resorting to biomedical interventions. The resulting delay in accessing biomedical care may exacerbate symptoms and lengthen the duration of untreated psychosis [61,62], and contributing to psychosis stigma within healthcare facilities. Healthcare facility staff may interact with a higher caseload of patients with severe psychotic symptoms [8], and which may impact their attitudes towards people with psychosis and have implications for diagnostic overshadowing and delayed diagnosis and treatment [63,64]. Specifically in the healthcare settings, participants highlighted the ways in which community stigma infiltrates psychosis care as many healthcare workers, particularly non-psychiatric and non-clinical staff, fail to reconcile community and personal beliefs regarding mental illness and their duty to provide high-quality care to PWLE. When presented with severe cases of psychosis, healthcare workers often held negative beliefs and prejudicial attitudes toward PWLE, likely due to fear and a lack of clinical psychiatric knowledge. Given the negative implications of stigma on health, well-being, and recovery, there is a clear need for integrating stigma-reduction activities into psychosis management both to help individuals cope with experiences of stigma as well as to build resilience and empower individuals to overcome internalized stigma. Investment in such programming has been limited in Africa; however non-pharmacological interventions in African countries have demonstrated reductions in harmful practices such as chaining as well as in stigma and discrimination [52,65–68]. As psychosis programming expands, interventions need to draw from evidence-based stigma-reduction principles to effectively support psychosis management.

Our study revealed a complex process of stigma within and around the family setting where family members may both perpetuate and receive stigma themselves. In settings with pervasive poverty, contributing to the family and community is vital. Pervasive beliefs around the capability of people with psychosis, reinforced by exclusion from development opportunities or employment discrimination, may drive poor treatment by family members. There is likely a the strong

inter-relationship between mental illness, poverty, and stigma [61]. Family members also face stigma due to their relative's psychosis and fear the consequences of disclosing their relative's diagnosis. This phenomenon has also been observed among families of people with psychosis in Ethiopia who perceived or experienced stigma and hid their relative's mental illness [62]. In settings such as Malawi where resources are limited and the burden of caregiving falls on the family, supportive programs for families attentive to stigma are warranted. Further, rehabilitation programs that support individuals with psychosis in generating income may be an important supportive care strategy that helps both to challenge inaccurate stereotypes about ability to contribute as well as to decrease stigma for both the individual and their family.

## Limitations

There are several inherent limitations to this study. With respect to positionality, while diverse, the study team consisted of biomedical health practitioners and researchers, with some members from the global north. Due to resource and time constraints, we were unable to involve PWLE in the interpretation of the data. While efforts were made to include diverse perspectives and experiences, our analytic lens may reflect a euro-centric, biomedical bias.

Considering our application of the Health Stigma and Discrimination Framework, the IDIs and FGDs yielded limited data around the role of laws and policies, media, law enforcement practices, access to justice, advocacy, or social protections in the stigmatization process and failed to investigate intersectionality. With an eye for dismantling the underlying power structures that drive health inequity and inequality, further inquiry into these topics may be warranted.

With respect to sampling, we included PWLE who were engaged in care at an urban outpatient psychiatric clinic (and their caregivers). Most (9/12) PWLE had low CGI scores (<3), indicating minimal or subthreshold psychotic symptoms. As such, the sample is not generalizable to more rural populations, where access to psychiatric care is more challenging, or to those with more severe symptoms who may not be engaged in care. While care was taken to ensure participants' comfort and anonymity, we sampled PWLE, caregivers and medical providers from a facility where they receive care or are employed. Thus, it is possible that participants' responses could be subject to social desirability biases. Finally, the religious leaders all belonged to a Christian denomination, while over 10% of the Malawian population are Muslim.

## Conclusion

This qualitative study featured a novel application of the Health Stigma and Discrimination framework to understand stigma among people with psychosis. Findings from this study detail key drivers of stigma, manifestations, outcomes and health and social impacts. In unpacking the stigmatization process, we identified areas for intervention to disrupt the initial causes of stigma, support PWLE and their family to cope with experiences of discrimination and better engage in care, and address the resulting health and social harms. Further, clinical care for patients with psychosis should be attentive to the negative impact of stigma. In partnership with PWLE, further research and investment in stigma reduction efforts, particularly in the integration of stigma reduction into existing psychosis management programs, are warranted.

## Author contributions

**Conceptualization:** Melissa A. Stockton, Brian W. Pence, Kazione Kulisewa.

**Formal analysis:** Melissa A. Stockton, Jack Kramer, Griffin Sansbury, Jackson Devadas.

**Funding acquisition:** Brian W. Pence.

**Methodology:** Melissa A. Stockton, Brian W. Pence, Kazione Kulisewa.

**Project administration:** Harriet Akello Tikhiwa.

**Resources:** Harriet Akello Tikhiwa.

**Supervision:** Melissa A. Stockton, Brian W. Pence, Kazione Kulisewa.

**Visualization:** Jack Kramer.

**Writing – original draft:** Melissa A. Stockton, Jack Kramer.

**Writing – review & editing:** Joshua Chienda, Abigail M. Morrison, Griffin Sansbury, Alex Zumazuma, Hillary Mortensen, Mwawi Ng'oma, Patrick Nyirongo, Isaac Mtonga, Bonginkosi Chiliza, Anthony Peter Sefasi, Patani Mhango, Bradley N. Gaynes, Brian W. Pence, Kazione Kulisewa.

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
