## [Decision Letter · Decision Letter 0]

19 Nov 2024

PMEN-D-24-00333

Applying the Health Stigma and Discrimination Framework to Psychosis Stigma in

Malawi

PLOS Mental Health

Dear Dr. Stockton,

Thank you for submitting your manuscript to PLOS Mental Health. After careful consideration, we feel that it has merit but does not fully meet PLOS Mental Health’s publication criteria as it currently stands. Therefore, we invite you to submit a revised version of the manuscript that addresses the points raised during the review process.

We look forward to receiving your revised manuscript.

Kind regards,

Bochra Nourhene Saguem, M.D.

Academic Editor

PLOS Mental Health

Journal Requirements:

1. Please include a complete copy of PLOS’ questionnaire on inclusivity in global research in your revised manuscript. Our policy for research in this area aims to improve transparency in the reporting of research performed outside of researchers’ own country or community. The policy applies to researchers who have travelled to a different country to conduct research, research with Indigenous populations or their lands, and research on cultural artefacts. The questionnaire can also be requested at the journal’s discretion for any other submissions, even if these conditions are not met.  Please find more information on the policy and a link to download a blank copy of the questionnaire here: https://journals.plos.org/plosmentalhealth/s/best-practices-in-research-reporting. Please upload a completed version of your questionnaire as Supporting Information when you resubmit your manuscript.

2. We ask that a manuscript source file is provided at Revision. Please upload your manuscript file as a .doc, .docx, .rtf or .tex.

3. Please provide separate figure files in .tif or .eps format.

https://journals.plos.org/mentalhealth/s/figures 

https://journals.plos.org/mentalhealth/s/figures#loc-file-requirements 

Additional Editor Comments (if provided):

Reviewers' comments:

Reviewer's Responses to Questions

**Comments to the Author**

1. Does this manuscript meet PLOS Mental Health’s publication criteria ? Is the manuscript technically sound, and do the data support the conclusions? The manuscript must describe methodologically and ethically rigorous research with conclusions that are appropriately drawn based on the data presented.

Reviewer #1: Yes

Reviewer #2: Yes

2. Has the statistical analysis been performed appropriately and rigorously?

Reviewer #1: Yes

Reviewer #2: N/A

3. Have the authors made all data underlying the findings in their manuscript fully available (please refer to the Data Availability Statement at the start of the manuscript PDF file)?

Reviewer #1: Yes

Reviewer #2: Yes

4. Is the manuscript presented in an intelligible fashion and written in standard English?

Reviewer #1: Yes

Reviewer #2: Yes

5. Review Comments to the Author

Reviewer #1: Thankyou for the opportunity to review this excellent paper. I have a couple suggestions for clarity and enrichment of your discussion but overall, feel it is a very interesting and important contribution to global mental health care

Introduction -

Your definitions of stigma are robust and draw on the work of Link & Phelan (and others) who are known scholars in this field. Given the allusion to Goffman's 'deeply discrediting attributes', I would suggest drawing on his work as well, as the seminal scholar on stigma.

The definition / elucidation of your chosen framework is fairly brief - it is novel, and a good application - I would recommend a little more detail here of the framework and your rationale for using it.

Data collection -

Your methodology appears confined to a single opening sentence here. For a qualitative approach, please expand on the constructivist framework and its relevance / influence on the research. Which theorists did you draw on?

Although not a requirement for rigour or trustworthiness, there feel a missed opportunity in your results/ discussion exploring the relationship between mental health care and the persisting community perceptions around witchcraft and spirituality as drivers in mental illness - a more detailed context of how MH care is enacted in this context would add depth to the descriptions of stigma. Perhaps a brief historical overview of those community beliefs, or a sense of how healthcare professional in Malawi mediate between care and community beliefs?

Thankyou again - a pleasure to review and such interesting research.

Reviewer #2: Important piece of research in the field of stigma and discrimination and Psychoses.

1. Abstract – Add “developed semi-structured qualitative guides based in a constructivist epistemology and formative research methodologies” to the methods section

2. Introduction

a. Any qualitative study require a reading of the context of understand the underlying mechanisms of the process of stigma and discrimination. The authors need to describe the context of Malawi in this section.

b. It will be useful to elaborate the reasons for choosing the Health Stigma and Discrimination framework – Page 8 lines 99-103

3. Sample recruited

a. What was the duration of illness of those with Psychoses? Was everyob=ne under treatment at the time of recruitment?

b. What was the relationship of the family caregivers of PWLE?

c. How was informed consent obtained by the researchers, especially when educational levels were low?

4. Results and the Framework figure are well presented. It will be helpful if the researches can document the frequency of responses. For example, how many of the recruited participants reported on fear?

Was there any response in the data set that did not fit the framework?

As an option – the authors may present a table depicting the components of the framework with quotes as examples for each component.

5. Discussion is fairly well written.

The authors could refer to the work of Koschorke et all in Social Science and medicine on work done in another LMIC - India (Koschorke M et al, . Experiences of stigma and discrimination of people with schizophrenia in India. Soc Sci Med. 2014 Dec;123:149-59. doi: 10.1016/j.socscimed.2014.10.035. Epub 2014 Oct 18. PMID: 25462616; PMCID: PMC4259492.)

6. PLOS authors have the option to publish the peer review history of their article (what does this mean? ). If published, this will include your full peer review and any attached files.

**Do you want your identity to be public for this peer review?** For information about this choice, including consent withdrawal, please see our Privacy Policy .

Reviewer #1: **Yes: ** Dr Floraidh Rolf

Reviewer #2: No

---

## [Decision Letter · Decision Letter 1]

25 Mar 2025

Applying the Health Stigma and Discrimination Framework to Psychosis Stigma in

Malawi

PMEN-D-24-00333R1

Dear Ms. Stockton,

We are pleased to inform you that your manuscript 'Applying the Health Stigma and Discrimination Framework to Psychosis Stigma in

Malawi' has been provisionally accepted for publication in PLOS Mental Health.

Best regards,

Pierre Olivier Jacquet, PhD

Academic Editor

PLOS Mental Health

Reviewer Comments (if any, and for reference):

Reviewer's Responses to Questions

**Comments to the Author**

1. If the authors have adequately addressed your comments raised in a previous round of review and you feel that this manuscript is now acceptable for publication, you may indicate that here to bypass the “Comments to the Author” section, enter your conflict of interest statement in the “Confidential to Editor” section, and submit your "Accept" recommendation.

Reviewer #1: All comments have been addressed

Reviewer #2: All comments have been addressed

Reviewer #3: All comments have been addressed

2. Does this manuscript meet PLOS Mental Health’s publication criteria ? Is the manuscript technically sound, and do the data support the conclusions? The manuscript must describe methodologically and ethically rigorous research with conclusions that are appropriately drawn based on the data presented.

Reviewer #1: Yes

Reviewer #2: Yes

Reviewer #3: Yes

3. Has the statistical analysis been performed appropriately and rigorously?

Reviewer #1: Yes

Reviewer #2: N/A

Reviewer #3: N/A

4. Have the authors made all data underlying the findings in their manuscript fully available (please refer to the Data Availability Statement at the start of the manuscript PDF file)?

Reviewer #1: Yes

Reviewer #2: No

Reviewer #3: Yes

5. Is the manuscript presented in an intelligible fashion and written in standard English?

Reviewer #1: Yes

Reviewer #2: Yes

Reviewer #3: Yes

6. Review Comments to the Author

Reviewer #1: Thankyou for your revision. This paper is excellent and ready for publication.

Reviewer #2: The purpose of seeking information on the frequency of responses was simply to get a sense of the frequency in relation to the total number interviewed. For example - Line 430-431, “I do not think that people would want to be close friends with someone that is living with this mental disorder” (PW-12). It is likely that majority of the participants - PWs - may have responded in a similar manner. This would indicate that this is a general opinion.

Otherwise - no comments

Reviewer #3: Excellent article and admire the way the issue of stigma has been addressed.

Especially relevant findings are the issue of internalised stigma and the impact of stigma in delayed seeking / receiving treatment.

7. PLOS authors have the option to publish the peer review history of their article (what does this mean? ). If published, this will include your full peer review and any attached files.

**Do you want your identity to be public for this peer review?** For information about this choice, including consent withdrawal, please see our Privacy Policy .

Reviewer #1: No

Reviewer #2: **Yes: ** Padmavati Ramachandran

Reviewer #3: **Yes: ** RAVINDRA NARAYAN AGRAWAL
